# Consequence of Histoincompatibility beyond GvH-Reaction in Cytomegalovirus Disease Associated with Allogeneic Hematopoietic Cell Transplantation: Change of Paradigm

**DOI:** 10.3390/v13081530

**Published:** 2021-08-03

**Authors:** Matthias J. Reddehase, Rafaela Holtappels, Niels A. W. Lemmermann

**Affiliations:** Institute for Virology and Research Center for Immunotherapy (FZI), University Medical Center of the Johannes Gutenberg-University Mainz, 55131 Mainz, Germany; r.holtappels@uni-mainz.de (R.H.); lemmermann@uni-mainz.de (N.A.W.L.)

**Keywords:** avidity, antigen presentation, CD8 T cells, cytomegalovirus, cytomegalovirus disease, graft-versus-host disease (GvHD), hematopoietic cell transplantation (HCT), hematopoietic reconstitution

## Abstract

Hematopoietic cell (HC) transplantation (HCT) is the last resort to cure hematopoietic malignancies that are refractory to standard therapies. Hematoablative treatment aims at wiping out tumor cells as completely as possible to avoid leukemia/lymphoma relapse. This treatment inevitably co-depletes cells of hematopoietic cell lineages, including differentiated cells that constitute the immune system. HCT reconstitutes hematopoiesis and thus, eventually, also antiviral effector cells. In cases of an unrelated donor, that is, in allogeneic HCT, HLA-matching is performed to minimize the risk of graft-versus-host reaction and disease (GvHR/D), but a mismatch in minor histocompatibility antigens (minor HAg) is unavoidable. The transient immunodeficiency in the period between hematoablative treatment and reconstitution by HCT gives latent cytomegalovirus (CMV) the chance to reactivate from latently infected donor HC or from latently infected organs of the recipient, or from both. Clinical experience shows that HLA and/or minor-HAg mismatches increase the risk of complications from CMV. Recent results challenge the widespread, though never proven, view of a mechanistic link between GvHR/D and CMV. Instead, new evidence suggests that histoincompatibility promotes CMV disease by inducing non-cognate transplantation tolerance that inhibits an efficient reconstitution of high-avidity CD8^+^ T cells capable of recognizing and resolving cytopathogenic tissue infection.

## 1. Introduction

Human cytomegalovirus (hCMV) is the prototype member of the β-subfamily of the herpes virus family [1]. Public awareness is low, because an intact immune system efficiently resolves primary infection. Infection can principally occur in all age groups, but the virus is often acquired in early childhood and may be viewed epidemiologically as a “daycare center infection” that usually passes with only mild, feverish symptoms. CMV is thus rarely considered in a differential diagnosis of age-typical, harmless infections [2]. Upon contact between the asymptomatically infected child and a CMV-naïve expectant mother in secondary pregnancy, there exists a risk of primary infection and dia-placental transmission [3], resulting in congenital infection of the immunological immature fetus with a range of birth defects collectively known as cytomegalic inclusion disease (CID) [2,4,5,6,7].

Other risk groups of significant clinical relevance are patients who undergo an immunocompromising therapy of unrelated primary diseases. This includes recipients of solid organ transplantation (SOT), who receive immunosuppressive treatment to prevent graft rejection by a host-versus-graft (HvG) reaction (for clinical overviews, see [5,8,9]). The focus of this brief review is on hematopoietic (stem) cell (HC) transplantation (HCT), which is the last therapeutic option to cure aggressive types of hematopoietic malignancies that are refractory to standard anti-tumor therapies. In essence, malignant cells are wiped out by hematoablative treatment, which, unavoidably, also depletes non-malignant hematopoietic cells of all differentiation lineages, including mature cells that mediate innate and adaptive immunity. HCT is the means to repopulate the bone marrow stroma with hematopoietic stem and progenitor cells and thus to reconstitute the immune system. The phase of transient immunodeficiency after HCT is a “window of opportunity” for hCMV to reactivate to productive infection within the latently infected donor HC or within latently infected cells residing in organs of the recipient, or both (reviewed and discussed in [10]). Lack of immune control leads to an unrestricted inter- and intra-tissue virus spread with histopathological lesions that can lead to multiple-organ CMV disease. Interstitial pneumonia represents the most feared disease manifestation with often lethal outcome, particularly when the clinical hCMV variant is resistant to common antiviral drugs [11,12,13]. In such cases, adoptive immunotherapy by transfer of antiviral CD8^+^ T cells is the last resort to close the “gap of risk” between hematoablative treatment of the primary disease and complete immune reconstitution by HCT [14,15,16,17,18,19,20,21]. Follow-up monitoring of HCT recipients by PCR to detect virus reactivation with high sensitivity is routine at transplantation centers worldwide to initiate antiviral therapies at the earliest possible moment, a strategy known as “pre-emptive” therapy [12].

It is longstanding clinical experience that HCT-associated CMV disease, when compared to syngeneic HCT with identical twins as donor and recipient [22,23] or even autologous HCT [24], is of higher incidence and often more severe when a family or unrelated HC donor and the recipient differ in major histocompatibility (MHC/HLA) or minor histocompatibility (minor-H) loci [24,25,26]. Such clinical settings are referred to as “allogeneic” HCT. Specifically, HCT with an HLA-identical sibling donor and recipient pair is an allogeneic HCT based on differences in minor-H loci [26]. As such immunogenetic mismatches are the basis for an immunological graft-versus-host (GvH) reaction and disease (GvHR/D) mediated primarily by donor T cells specific for non-shared histocompatibility antigens, the donor and recipients are HLA type-matched as close as practicable for risk management. Despite this, differences in minor-H antigens (minor-HAg) are unavoidable and bear a risk of GvHD. Stern and colleagues [27] recently reviewed the issue of hCMV latency and reactivation in recipients of allogeneic HCT. Here we focus on the specific question of whether post-HCT GvH/D and CMV disease are mechanistically linked, or if both are independent consequences of the underlying histoincompatibility.

In a superficial view, immunogenetic mismatches and GvHR/D are used as if they were synonyms. Clinical treatment regimens leave an uncertainty, however, if GvHR/D after allogeneic HCT is caused by effector functions of host-reactive donor-type effector cells derived from lymphoid-lineage hematopoietic differentiation and thymic education, or rather from mature donor T cells already present in the HC transplant. In fact, the HC transplant is often deliberately left undepleted of mature T cells, accepting adverse effects of GvHR to maintain a beneficial graft-versus-leukemia/lymphoma (GvL) reaction for reducing the risk of tumor relapse from minimal residual leukemia/lymphoma. This, next to GvHD and CMV disease, is a major, if not the major, concern in tumor therapy by HCT. Accordingly, separating GvH-reactive from GvL-reactive cells is a topic of intense research ([28,29], reviewed in [30]). In addition, in the case of a CMV^+^ donor, mature antiviral T cells in the HC transplant can also exert a beneficial graft-versus-infection (GvI) effect [31]. Therefore, the precise and highly individualized regimen of allogeneic HCT always demands a benefit–risk assessment depending on the individual donor–recipient constellation. In fact, hematopoietic reconstitution originating from allogeneic hematopoietic stem cells is expected to lead to “transplantation tolerance”. A conditioning HCT performed with HC from an SOT donor has even been discussed as a potential strategy to prevent graft rejection by tolerizing SOT recipients against MHC/HLA antigens and/or minor-HAg expressed by the donor tissue (for an overview, see [32]).

Despite all undeniable genetic differences between the host species adapted human and animal CMVs, as well as between their respective hosts, the mouse model has, in the past, already proven its validity for human CMV disease by identifying fundamental, CMV-common principles of pathogenesis, immune control, and therapeutic intervention (reviewed in [33]). We have explained this with biological convergence during virus–host co-evolution [33]. Specifically, immunotherapy of post-HCT CMV disease by adoptive transfer of CD8^+^ T cells in the mouse model based on murine CMV (mCMV) ([34,35], reviewed in [36,37,38]) has been successfully translated to preemptive immunotherapy of CMV disease in clinical HCT settings (see above). In the weakness of any reductionistic approach in animal models to never reproduce the clinical reality in all its complexity lies also the strength of defined conditions. In fact, human CMV disease cannot be fully reproduced by any model because it is never a predictable, uniform entity. It depends on the individual’s genetic constitution and infection history that defines the latent CMV genome load and incidence of reactivation (discussed in [39]), and on genetic differences in hCMV variants/strains. These likely differ between donor and recipient and can have a fundamental impact on cell-type tropism and thus on pathogenicity, as well as on immunological marks in terms of antigens and immune evasion proteins expressed [1,40,41]. In addition, the primary malignancy and its treatment history are also determinants for the outcome of HCT in clinical real life. Thus, no model will ever perfectly suit human CMV disease in any individual HCT recipient.

Here we summarize and interpret two recent reports on CMV disease in mouse models of allogeneic HCT performed in an immunogenetic GvH transplantation direction across a single MHC class-I difference [42], as well as across a difference in a minor-H locus [43]. Combined, both models led us to the conclusion that lethal CMV disease occurs despite the absence of GvH-reactive effector cells and, instead, can be caused by non-cognate transplantation tolerance that inhibits the efficient reconstitution of antiviral CD8^+^ T cells, thereby leading to unrestricted virus spread and histopathology.

## 2. Key Results from Mouse Models of Allogeneic HCT

### 2.1. Lethality from CMV Infection after HCT in Immunogenetic GvH Transplantation Direction

The impact of mCMV infection on the outcome of syngeneic experimental HCT, with sex-matched BALB/c mice as donors and recipients, has been studied extensively (for reviews, see [37,38]). In this model, only epigenetic differences between individual mice might induce GvHR/D, which is not the case only in autologous HCT. In essence, control of mCMV infection and prevention of lethality was found to depend on the number of transplanted HCs and correlated with the efficient and timely reconstitution of high-avidity antiviral CD8^+^ T cells [37,38].

A more recently studied model of allogeneic HCT across an MHC class-I disparity used BALB/c mice and mutant BALB/c-H-2^dm2^ mice alternatingly as HCT donors and recipients [42]. BALB/c mice express the full set of MHC class-I molecules of the *H-2^d^* haplotype, namely K^d^, D^d^, and L^d^. Based on a spontaneous genetic deletion, the L^d^ molecule is absent in the otherwise genetically identical strain BALB/c-H-2^dm2^. This special feature provided the chance to perform HCT selectively as either HvG-HCT or GvH-HCT. Theoretically, the transplantation direction in HvG-HCT allows only a unidirectional response of recipient–resident CD8^+^ T cells against the L^d^ molecule expressed on transplanted donor-derived HC, whereas the transplantation direction in GvH-HCT allows only a unidirectional response of reconstituted donor-genotype CD8^+^ T cells against the L^d^ molecule expressed on all cell types of the recipient (Figure 1).

The immunogenetic direction made a fundamental difference in the outcome of HCT. Whereas most recipients of HvG-HCT survived the infection, none of the recipients survived under conditions of GvH-HCT (Figure 1A). As clearly shown for the liver, survival of HvG-HCT recipients correlates with tissue infiltration by T cells that cluster around the few remaining infected hepatocytes, thereby forming so-called “nodular inflammatory foci” (NIF) confining, and eventually resolving, productive infection. In sharp contrast, infiltrating T cells are scarce and NIF are barely formed after GvH-HCT. As a consequence of missing immune control, the infection spreads unhindered, resulting in extended viral histopathology (Figure 1B). Notably, evidence for GvHD-typical histopathology is missing.

### 2.2. Failure in the Reconstitution of High Avidity Virus-Specific CD8^+^ T Cells after GvH-HCT 

For explaining the selective failure of antiviral control in GvH-HCT recipients compared to efficient antiviral control in HvG-HCT recipients, CD8^+^ T cells were isolated from the livers of both groups of HCT recipients. Viral epitope-specific IFNγ^+^CD8^+^ T cells were quantitated depending on their functional avidity in recognizing cell surface peptide-MHC class-I complexes m164-D^d^, M105-K^d^, and m145-K^d^ (Figure 2) [42]. Responding cells were categorized into non-protective “low avidity” cells and protective “high avidity” cells based on the previous finding that an avidity corresponding to an exogenous peptide loading concentration of ≤10^−9^ M is needed for the recognition of antigenic peptides presented by infected cells following endogenous antigen processing [36,44]. At a glance, compared to HvG-HCT, frequencies of viral epitope-specific cells were generally low, and were even lower in the high-avidity compartment, after GvH-HCT. This result identified an insufficient reconstitution of antiviral CD8^+^ T cells as the reason for unrestricted virus spread and extensive viral histopathology after GvH-HCT. 

### 2.3. Enhancement of Antigen Presentation Restores Antiviral Protection after GvH-HCT

A second model of T cell-depleted allogeneic HCT used C57BL/6 (*H-2^b^* haplotype) mice as donors and BALB.B mice as recipients in a GvH-HCT [43]. These two mouse strains share the major histocompatibility complex and are thus identical in MHC class-I as well as class-II antigens, while differing in genetic background, including minor histocompatibility loci. A well-studied and particularly strong minor-HAg is H60 (for a review, see [45]), which is not expressed in C57BL/6 donors of GvH-HCT but is expressed in the BALB.B recipients (Figure 3A).

In the context of mCMV infection, it is worth noting that H60 is an activatory ligand of the natural killer (NK) cell receptor NKG2D, and is downregulated in infected cells by the viral m155 gene product [46]. As NKG2D is expressed also on activated CD8^+^ T cells, serving as a costimulatory receptor, H60-NKG2D ligation could even enhance GvH-reactivity against H60-expressing uninfected cells. 

Based on the experience of poor reconstitution of high-avidity epitope-specific CD8^+^ T cells in the MHC class-I mismatch model of GvH-HCT ([42], see above), BALB.B recipients were infected either with wild-type (WT) virus or with a mutant deleted in viral genes that encode immunoevasive regulators of antigen presentation (ΔvRAP).

The idea behind infection with ΔvRAP virus was to relieve peptide-MHC class-I complexes of the vRAP-mediated inhibition of their transport to the cell surface, and thus to enhance antigen presentation for recruiting also low-avidity virus-specific CD8^+^ T cells into NIF for the recognition of infected cells ([42,43] and referencing of vRAP functions therein). As was the case in the MHC class-I mismatch model (recall Figure 1), GvH-HCT was associated with significant lethality from infection with WT virus expressing vRAPs that inhibit antigen presentation. In contrast, despite a genetic predisposition to GvHR directed against the minor-HAg H60, all recipients survived when antigen presentation was not inhibited by vRAP (Figure 3A). Notably, there was no evidence for a GvHR occurring at all in the BALB.B recipients regardless of the type of infecting virus, as CD8^+^ T cells specific for the H60-derived antigenic peptide LTFNYRNL (also known as LYL8) were not detected [43]. This indicated an absence of GvHR/D due to the establishment of transplantation tolerance (for reviews, see [32,47]).

The histopathological correlate of lethality from WT virus infection of the minor-HAg H60^+^ GvH-HCT recipients is the overall scarcity of liver tissue-infiltrating T cells, missing NIF formation, and extensive viral spread, whereas enhanced presentation of viral peptides after infection with the ΔvRAP virus restored T-cell infiltration, NIF formation, and clearance of productive infection, thus avoiding extensive viral histopathology (Figure 3B).

These findings are decisive for revealing the mechanism that causes lethality in the GvH-HCT infection model. Enhancement of viral antigen presentation in infected cells by vRAP deletion can modulate only the recognition of infected cells by effector T cells, and thus can prevent only viral histopathology. It cannot, however, prevent histopathology resulting from a GvH T-cell attack against uninfected tissue cells. These data formally exclude lethal histopathology as resulting from GvHR/D and, instead, provide strong evidence for death being caused by viral histopathology and consequent organ failure.

## 3. Summary

In GvH-HCT and infection with WT virus, reconstitution of viral peptide-specific CD8^+^ T cells is generally at a low level, and the few cells present are mostly of a functional avidity that is not high enough for recognizing infected cells, in which the presentation of antigenic peptides is limited through the action of vRAP. As a result, unhindered virus cell-to-cell spread can lead to extensive tissue lesions (Figure 4, left panel).

In HvG-HCT and, accordingly, in syngeneic HCT, after infection with WT virus, reconstitution of viral peptide-specific CD8^+^ T cells is generally at an elevated level, and many cells are of a functional avidity that is high enough to recognize infected cells under conditions of limited peptide presentation. Moreover, once sensitized by TCR signaling, the high-avidity cells release IFNγ, which is known to oppose vRAP function [48]. This enhances antigen presentation and recruits even low-avidity CD8^+^ T cells to infected cells for recognition. Combined, these mechanisms lead to NIF formation, prevention of viral spread, and, eventually, to resolution of productive infection (Figure 4, center panel).

In GvH-HCT and infection with ΔvRAP virus, elevated presentation of antigenic peptides can recruit even low-avidity CD8^+^ T cells into NIF for the recognition of infected cells, prevention of viral spread, and termination of productive infection (Figure 4, right panel).

## 4. Conclusions

The data provide reasonable evidence to conclude that lethality associated with CMV infection in recipients of allogeneic HCT is not caused by virus-mediated aggravation of GvHD. Instead, it results from enhanced viral pathogenesis that reflects a failure in the control of infection. Absence of protective antiviral CD8^+^ T cells does not result from effector functions of GvH-reactive T cells. Rather, the failure in antiviral control can be attributed to an inefficient lympho-hematopoietic reconstitution of high-avidity antiviral CD8^+^ T cells capable of recognizing limited antigen presentation by infected cells. As the mechanism of inefficient reconstitution of protective antiviral immunity, we propose that transplantation tolerance against MHC or minor-H mismatch-antigens leads to a bystander, “non-cognate tolerance” against viral antigens.

## 5. Lesson Learned for Better Clinical Understanding

In accordance with the two mouse models of allogeneic HCT discussed here, clinical studies have revealed that a quantitative loss of hCMV-specific CD8^+^ T cells underlies the uncontrolled virus replication after allogeneic HCT [49,50]. As discussed above (see the introduction), no experimental animal model can cover human disease in all its complexity with broad variance between individual recipients of allogeneic HCT and infection with mostly uncharacterized virus variants. Obviously, in cases of a canonical GvHR/D, particularly when mature donor T cells are not depleted to maintain a GvL effect for preventing leukemia relapse, elimination of antigen presenting cells of the recipient necessarily interferes with the reconstitution of protective antiviral T cells. Conversely, immunosuppressive treatment for GvHR/D prophylaxis also prevents, or at least critically delays, the reconstitution of antiviral immunity.

Reducing the complexity and numbers of variables could be seen as a limitation of experimental animal models, but it offers also the possibility of dissecting different mechanisms that have a similar outcome and are thus difficult to distinguish by clinical investigation. These two mouse models of allogeneic HCT contribute the important finding that mismatches in major or minor histocompatibility antigens result in a failure of antiviral CD8^+^ T-cell reconstitution, independent of GvHR/D.

This new insight into the mechanism of CMV disease after allogeneic HCT gives a further argument for immunotherapy by adoptive transfer of antiviral CD8^+^ T cells, ideally not by short-lived effector cells but by long-lived memory cells, to reconstitute antiviral immunity enduringly. We, like many others, previously thought and argued that immunotherapy serves, primarily, to bridge the “window of risk” in a transient phase of immunodeficiency, until endogenous lympho-hematopoietic reconstitution of antiviral CD8^+^ T cells takes over. Now we are faced with the possibility that, under conditions when allogeneic HCT and CMV infection coincide, transplantation tolerance towards the antigenic mismatch leads to a lasting “non-cognate tolerance” against CMV antigens. The situation is reminiscent of the idea of using allogeneic HCT for inducing tolerance towards a subsequent solid organ allograft from the same donor (reviewed in [32]), though with the difference that non-cognate tolerance towards viral antigens is adverse in its consequences. We propose that this mechanism might leave recipients, who recovered from allogeneic HCT, in a state of long-lasting CMV-selective immunodeficiency.

## Figures and Tables

**Figure 1 viruses-13-01530-f001:**
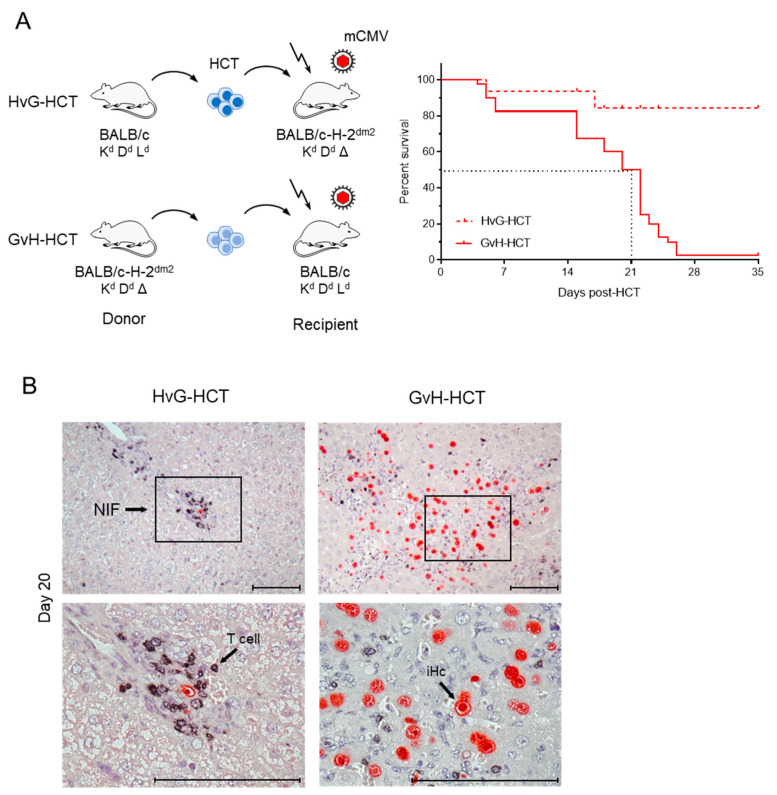
Comparison of lethality and viral histopathology in mCMV-infected recipients of HCT performed in immunogenetic HvG or GvH direction. (**A**) Model and Kaplan-Meier survival curves. The flash symbol indicates hematoablative total-body γ-irradiation. The dotted lines mark the 50% survival time. (**B**) 2-color immunohistological images of liver tissue sections (upper panels, overview; lower panels, resolved to greater detail) showing tissue infiltration by T cells (black staining) and infected hepatocytes (iHC, red staining). Frames demarcate regions resolved to greater detail in the corresponding lower panel images. NIF, nodular inflammatory focus. Bar markers: 100 μm. Reproduced from reference [42] in a new arrangement.

**Figure 2 viruses-13-01530-f002:**
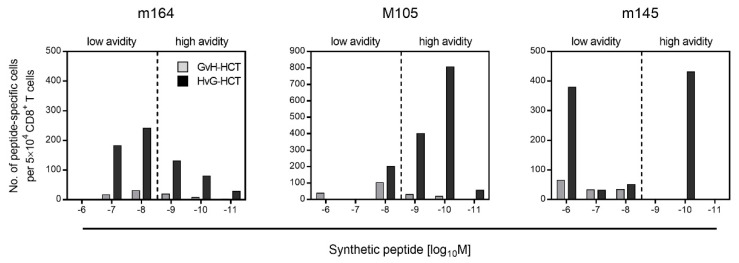
Frequencies and functional avidities of liver-derived, viral epitope-specific IFNγ^+^CD8^+^ T cells after GvH-HCT compared to HvG-HCT. Reproduced from reference [42] in a modified graphical presentation.

**Figure 3 viruses-13-01530-f003:**
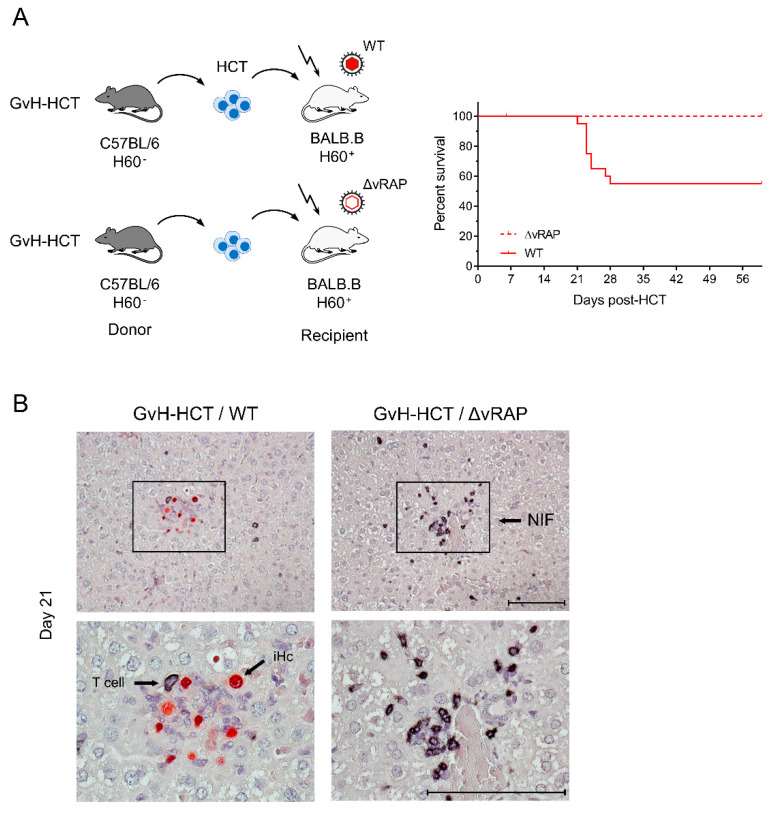
Comparison of lethality and viral histopathology in minor-HAg H60-mismatched GvH-HCT recipients infected with wild-type (WT) mCMV or a mutant of mCMV deleted for the genes that code for immunoevasive “viral regulators of antigen presentation” (ΔvRAP). (**A**) Model and Kaplan–Meier survival curves. The flash symbol indicates hematoablative total-body γ-irradiation. (**B**) 2-color immunohistological images of liver tissue sections (upper panels, overview; lower panels, resolved to greater detail) showing tissue infiltration by T cells (black staining) and infected hepatocytes (iHC, red staining). Frames demarcate regions resolved to greater detail in the corresponding lower panel images. NIF, nodular inflammatory focus. Bar markers: 100 μm. Reproduced from reference [43] in new arrangement.

**Figure 4 viruses-13-01530-f004:**
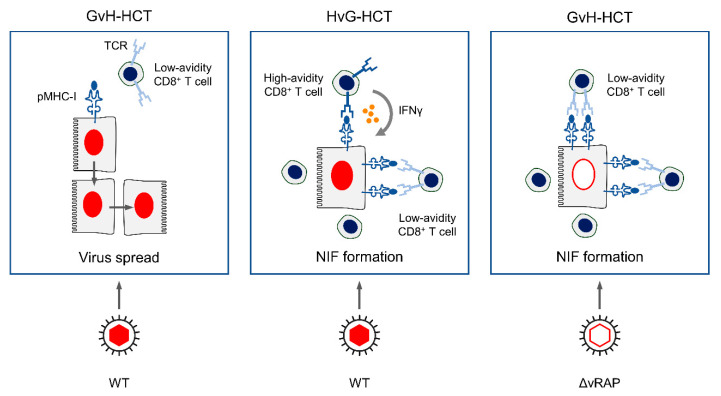
Graphical abstract. pMHC-I, peptide-loaded MHC-I molecules. TCR, T cell-receptor. NIF, nodular inflammatory focus, a microanatomical structure formed by CD8^+^ T cells that recognize infected tissue cells, here represented by hepatocytes. WT, wild-type mCMV. ΔvRAP, mCMV with deletion of genes coding for viral regulators of antigen presentation (vRAP) that inhibit cell surface trafficking of pMHC-I complexes. Adapted from reference [42].

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
