# Peer review of "Consequence of Histoincompatibility beyond GvH-Reaction in Cytomegalovirus Disease Associated with Allogeneic Hematopoietic Cell Transplantation: Change of Paradigm"

_viruses, 2021, doi:10.3390/v13081530_

Round 1

Reviewer 1 Report

The authors present an interesting summary on a potential mechanism of loss of viral control towards CMV in allogeneic HSCT. While the issue is of interest, some aspects need to be added especially in the discussion part. In the introduction part the authors focused in the depletion of stem cells and regarded the depletion of lymphocytes as secondary. In fact, for clinical allogeneic hematopoietic stem cell transplantation the other way around is true - HSCT can only be performed if the host is sufficiently immunosuppressed while the depletion of stem cells is less relevant especially in non-malignant diseases. Another issue is, that the authors focus on one mechanism but do not mention other potential reasons for failure of immune control of CMV in the alloHSCT setting.  Clinically, the problem is dominated by lack of CMV specific memory T cells and in the HLA-mismatch setting more intense immunosuppression (GVHD prophylaxis) adds together with mismatch of the CMV specific memory T cells from the donor with HLA mismatched HLA-molecules. Clinically, on the long run this problem usually resolves as soon new CMV specific donor T cells are induced which can be nicely shown also in the context of passive transfer of CMV specific T cells (third party or donor derived) which not only expand themself but during the course new clones emerge which have not been transferred passively. Another mechanism which contributes to the loss of control in a GVH setting which can also happen without clinical symptoms is depletion of host APCs. Especially in the HLA-mismatch setting it has been shown, that absence of GVHD but GVL effect is mediated by kill of host APC preventing alloreactivity via depletion of the cells needed for direct antigen presentation. While this mechanism protects from GVHD it may also contribute to less efficient control of viral infections. This problem usually resolves on the long run too since the APC are gradually replaced by donor APC. Still, this may be not as efficient as host APC in induction of CMV specific T cells since this situation permits indirect antigen presentation only. All these aspects to not diminish the value of the article which adds another mechanism but the article is currently too much focused on just one aspect and mentioning the additional mechanisms may improve the article.

Author Response

We thank the reviewer for alerting us of clinical aspects, and in particular of alternative mechanisms that can interfere with reconstitution in the allo-HCT patient. We are certainly aware of the fact that animal models are limited by not covering all variables that play a role in the clinical situation. We think we have made this clear quite frankly already in the Introduction (page 3, 2nd §), but we also agree that a reader expects a discussion of alternative - or let’s better say additional - mechanisms in the discussion. We have therefore extended the final Section 5 on page 9 (Lesson Learned for Better Clinical Understanding) by new §1 and §2.

Reviewer 2 Report

The Brief Review by Reddehase, Holtappels and Lemmermann, dealing with graft versus host reaction after hematopoietic stem cell transplantation (HCT), discusses the matter as to whether lethal CMV disease in this context is dependent on, or induces, graft versus host disease (GvHD). The review is clearly written, the topic properly introduced. The review is based on two previously published studies by the group, namely Holtappels et al., Front. Cell. Infect. Microbiol. 2020 and Gezinir et al., Front. Cell. Infect. Microbiol. 2020. In these publications CD8+ T-cell control of mouse cytomegalovirus infection in HCT mice was studied. Here, the authors draw conclusions from their combined observations.

In their first study they compare reactions in GvH and HvG using mice differing in only one MHC class I locus, Ld (with the other two MHC class I being identical). Their interesting finding is that in the HvG situation (recipient is lacking Ld) high avidity CD8+ T-cells protect the recipient, whereas in GvH situation (donor is lacking Ld) protective antiviral CD8+ T-cells were not reconstituted. This resulted in uncontrolled virus replication in a host showing no signs of GvH typical disease.

In the second GvH setting MHC class I loci were identical between donor and recipient, but the mice had a mismatch in the minor-H locus H60. Under these conditions wild-type MCMV infection was not controlled and again no signs of GvHD were observed. In the absence of MCMV encoded MHC class I inhibitors virus specific CD8+ T-cells was induced and the infection was controlled. H60 is a ligand for the activating NKG2D receptor expressed on NK cells and activated CD8+ T-cells and it is regulated by the MCMV m155 gene product. Even though this might not play a role for the graft versus infection reaction discussed here, it should be mentioned by the authors.

The conclusion the authors draw from their studies is that in GvH-HCT mice high avidity virus specific T-cells are not formed. The authors suggest that transplantation tolerance towards mismatched MHC and antigens leads to tolerance also against viral antigens. For sake of completeness, in addition to HvG-HCT, a setting using genetically identical mice and wild-type MCMV infection should be mentioned. Here, the situation of tolerance against mismatched MHC or minor-H loci would be absent, and the high avidity CD8+ T-cells should be reconstituted.

Altogether the authors present an intriguing model, resulting in many more questions that need to be answered in the future.

Author Response

In the second GvH setting MHC class I loci were identical between donor and recipient, but the mice had a mismatch in the minor-H locus H60. Under these conditions wild-type MCMV infection was not controlled and again no signs of GvHD were observed. In the absence of MCMV encoded MHC class I inhibitors virus specific CD8+ T-cells was induced and the infection was controlled. H60 is a ligand for the activating NKG2D receptor expressed on NK cells and activated CD8+ T-cells and it is regulated by the MCMV m155 gene product. Even though this might not play a role for the graft versus infection reaction discussed here, it should be mentioned by the authors.

We thank the reviewer for suggesting roles of H60 in addition to representing a minor-HAg. See new §2 and Reference 46 on page 6

The conclusion the authors draw from their studies is that in GvH-HCT mice high avidity virus specific T-cells are not formed. The authors suggest that transplantation tolerance towards mismatched MHC and antigens leads to tolerance also against viral antigens. For sake of completeness, in addition to HvG-HCT, a setting using genetically identical mice and wild-type MCMV infection should be mentioned. Here, the situation of tolerance against mismatched MHC or minor-H loci would be absent, and the high avidity CD8+ T-cells should be reconstituted.

As we have studied the syngeneic model for 2 decades, we forgot that the information may be important for a reader who may not have followed our work. We have now added this information on the last § on page 3.

Round 2

Reviewer 1 Report

The authors addressed all comments - I have no further issues